# ERAP1 and ERAP2 Enzymes: A Protective Shield for RAS against COVID-19?

**DOI:** 10.3390/ijms22041705

**Published:** 2021-02-08

**Authors:** Silvia D’Amico, Patrizia Tempora, Valeria Lucarini, Ombretta Melaiu, Stefania Gaspari, Mattia Algeri, Doriana Fruci

**Affiliations:** Department of Paediatric Haematology/Oncology and of Cell and Gene Therapy, Ospedale Pediatrico Bambino Gesù, IRCCS, 00146 Rome, Italy; silvia.damico@opbg.net (S.D.); patrizia.tempora@opbg.net (P.T.); valeria.lucarini@opbg.net (V.L.); ombretta.melaiu@opbg.net (O.M.); stefania.gaspari@opbg.net (S.G.); mattia.algeri@opbg.net (M.A.)

**Keywords:** COVID-19, renin-angiotensin system, ERAP1, ERAP2, risk factor

## Abstract

Patients with coronavirus disease 2019 (COVID-19) have a wide variety of clinical outcomes ranging from asymptomatic to severe respiratory syndrome that can progress to life-threatening lung lesions. The identification of prognostic factors can help to improve the risk stratification of patients by promptly defining for each the most effective therapy to resolve the disease. The etiological agent causing COVID-19 is a new coronavirus named severe acute respiratory syndrome coronavirus 2 (SARS-CoV-2) that enters cells via the ACE2 receptor. SARS-CoV-2 infection causes a reduction in ACE2 levels, leading to an imbalance in the renin-angiotensin system (RAS), and consequently, in blood pressure and systemic vascular resistance. ERAP1 and ERAP2 are two RAS regulators and key components of MHC class I antigen processing. Their polymorphisms have been associated with autoimmune and inflammatory conditions, hypertension, and cancer. Based on their involvement in the RAS, we believe that the dysfunctional status of ERAP1 and ERAP2 enzymes may exacerbate the effect of SARS-CoV-2 infection, aggravating the symptomatology and clinical outcome of the disease. In this review, we discuss this hypothesis.

## 1. Introduction

Infection with severe acute respiratory syndrome coronavirus 2 (SARS-CoV-2) has been identified as the cause of the coronavirus disease 2019 (COVID-19), which is currently affecting more than 100 million people globally, with more than 2 million deaths recorded by WHO as of 1 February 2021. SARS-CoV2 is one of seven CoVs known to infect human hosts in the upper respiratory tract or remain asymptomatic during the early stages of infection and evolve in some cases to multi-organ dysfunction and death (Figure 1) [1,2]. SARS-CoV, Middle East respiratory syndrome (MERS)-CoV, and SARS-CoV-2 belong to the latter group of CoVs. Following SARS-CoV-2 infection, patients experience a wide variety of clinical outcomes ranging from asymptomatic disease to respiratory syndrome and even death. Notably, a male bias in COVID-19 mortality has been described [3], with 60% of those dying from COVID-19 being male [4]. Interestingly, an analysis performed on 98 patients showed increased levels of the proinflammatory cytokines IL-8, IL-18, and CCL5 and reduced CD8^+^ T cell responses in males compared to females [4]. Notably, T-cell response was significantly and negatively correlated with patients’ age in males, but not females, suggesting differences in baseline immune capability during the early phase of SARS-CoV-2 infection [4,5]. A genetic predisposition could represent one of the discriminating elements underlying this different clinical outcome.

Based on genome sequencing, SARS-CoV-2 is about 82% identical to human SARS-CoV and 50% identical to MERS-CoV. However, while the latter uses dipeptidyl peptidase 4 (DPP4) as the primary receptor to enter the host cells, the entry of SARS-CoV and SARS-CoV-2 is mediated by the interaction between viral spike proteins and the extracellular domains of the angiotensin-converting enzyme 2 (ACE2) [6,7]. Although the spike proteins are 76.5% identical, the binding affinity of ACE2 for SARS-CoV-2 is 10–20-fold higher than that for SARS-CoV [7,8], thus explaining the greater transmissibility of SARS-CoV-2 compared to SARS-CoV [9]. Direct interaction of the CD147 receptor with the SARS-CoV-2 spike protein has also been proposed to mediate virion entry into host cells via endocytosis [10]. Because the results for this second type of receptor have not been confirmed by other authors [11], we will focus on ACE2 in this review.

ACE2 is a type I transmembrane protein with a catalytically active monocarboxypeptidase ectodomain that hydrolyzes various substrates, including angiotensin I (Ang I) and Ang II (Figure 2). It is widely expressed in several tissues, including nasal and oropharyngeal epithelium, where SARS-CoV-2 entrance occurs. The regulation of Ang II levels in the bloodstream is one of the most notable mechanisms for maintaining normotension and is strictly dependent on the balance between its synthesis and its metabolism [12]. ACE2 is an endogenous counter-regulator of the two main arms of the renin-angiotensin system (RAS): the classical one consisting of the ACE/Ang II/Ang II receptor type 1 (AT1R) and the alternative arm with the ACE2/Ang-(1-7)/Mas receptor (MasR). In the classical RAS axis, Ang II, the product of Ang I conversion, binds its AT1R receptor, increasing blood pressure and causing vasoconstriction and sodium retention (Figure 2a). Activation of this axis is known to induce inflammation, fibrosis, cellular growth, and vasoconstriction. In the second axis, ACE2 induces the conversion of Ang I to Ang-(1-9) and Ang II to Ang-(1-7). Ang-(1-7) interacting with MasR and AT2R opposes the action of the ACE-dependent pathway by causing vasodilation and reduced blood pressure. Via Ang II cleavage, ACE2 limits the availability of substrate in the potentially harmful ACE/Ang II/AT1R axis. At the same time, the production of Ang-(1-7) contributes toward stimulating the protective axis ACE2/Ang-(1-7)/MasR (Figure 2a) [13,14].

In addition to ACE and ACE2, the endoplasmic reticulum (ER) aminopeptidases, ERAP1 and ERAP2, are also involved in the RAS (Figure 2) [15,16,17]. ERAP1 and ERAP2 belong to the M1 zinc metallopeptidases with which they share the HEXXH(X)18E zinc-binding and GAMEN substrate recognition sequences that are essential for their enzymatic activity [18]. Although mainly localized in the ER, ERAP1 and ERAP2 can be released outside of cells upon stimulation with inflammatory cytokines and depending on the redox potential [15,19]. Based on their function in trimming amino acid residues at the N terminus of different substrates, ERAP1 and ERAP2 take part in multiple biological processes [20,21].

These enzymes are mainly known for their role in the ER via generating antigenic peptides with the correct length to bind MHC class I molecules [22]. Once formed, peptide-MHC class I complexes are transported on the cell surface to be recognized by cytotoxic CD8^+^ T cells and NK cells [18]. ERAP-mediated peptide trimming represents the last step of the antigen-processing pathway, an important mechanism that is initiated in the cytoplasm by the proteasome complex and other peptidases, which allows the immune system to recognize and eliminate virus-infected or transformed cells [23]. Like most of the components of the antigen-processing pathway, ERAP1 and ERAP2 are induced by inflammatory cytokines, such as IFNγ and TNFα [24,25]. Following stimulation with IFNγ and LPS, macrophages secrete ERAP1 and ERAP2, which help to enhance their phagocytic activity and nitric oxide (NO) synthesis [19,26,27]. The overexpression of ERAP1 in peripheral blood mononuclear cells (PBMC) results in the production of inflammatory cytokines and chemokines [28]. Specifically, ERAP1-treated PBMC showed the strongest production of IL-1β, IL-6, and TNFα, thus suggesting a pro-inflammatory role [28].

ERAP1 and ERAP2 digest plasma Ang II into Ang III and Ang IV [15,16,17], the major ligands of Ang type 2 receptor (AT2R) and Ang type 4 receptor (AT4R), respectively [29,30] (Figure 2). The interaction of Ang III and Ang IV with their receptors results in the reduction of inflammation and an increase in the production of NO, which contributes to the decrease in blood pressure [30]. The secretion of ERAP1 into the bloodstream is controlled by the interaction with ERp44, an ER resident thioredoxin-like-motif-containing protein, in a redox-dependent manner [15]. This phenomenon avoids the excessive release of ERAP1 into the bloodstream, preventing unfavorable hypotension. Currently, no evidence of direct interaction between ERAP2 and ERp44 has been demonstrated. Hypertension and increased circulating Ang II levels are often associated with the enhanced generation of vascular reactive oxygen species (ROS) [31], which are known to affect the ER redox potential [32] and induce the release of ERAP1 into the bloodstream [15]. Although ERAP enzymes have been extensively studied in the immunological context, their involvement in Ang II digestion and blood pressure regulation is indicative of a link to RAS that deserves further investigation.

## 2. *ERAP1* and *ERAP2* Gene Polymorphisms Associated with Hypertension

*ERAP1* and *ERAP2* genes are highly polymorphic. Dysfunctional genetic variants have been linked to the onset of autoimmune disease and viral infections [33,34,35,36,37]. Interestingly, most of these variants are also associated with essential hypertension and treatment responses in patients with left ventricular hypertrophy [38,39,40,41,42,43], further supporting their role in regulating blood pressure and cardiac remodeling (Table 1).

Zee and colleagues evaluated the potential association of 33 *ERAP1* and 12 *ERAP2* single nucleotide polymorphisms (SNPs) with blood pressure progression and incident hypertension in a cohort of 17,255 initially healthy White U.S. women (Table 1) [39]. The authors found three SNPs (*ERAP1* rs469783 and rs10050860; *ERAP2* rs2927615) to be associated with the risk of incident hypertension and one variant (*ERAP1* rs27772) associated with blood pressure progression [39]. Interestingly, the exposure of endothelial cells to Ang II significantly increased ERAP1, suggesting that AT1R activation through Ang II could induce *ERAP1* expression as a compensatory response [39]. In another study, two other SNPs of the *ERAP1* gene in the 3′UTR region, rs27980 and rs17086651, were associated with essential hypertension in a cohort of northeast Han Chinese patients (Table 1) [40]. Moreover, the *ERAP1* rs30187 loss of function variant (K528R) was linked to the reduced degradation of Ang II and essential hypertension in a cohort of 143 hypertensive and 348 normotensive Japanese subjects (Table 1) [41,42]. Ranjit and colleagues showed a significant association between the *ERAP1* rs30187 variant, volume expansion, and increased systolic and diastolic blood pressure in a cohort of 435 patients involved in the Hypertensive Pathotype (HyperPATH) Consortium, with stronger evidence in males [43]. Nevertheless, men bearing the rs30187 allele displayed elevated aldosterone (ALDO) levels with normal renovascular function as compared to women [43]. Similar results were obtained from *ERAP1*^+/−^ mice, which display significantly higher levels of blood pressure and Ang II in heart, kidney, and aorta tissues compared to wild-type mice [43]. Interestingly, this condition was associated with increased ALDO levels in males, but not females, thus indicating the existence of different mechanisms between the sexes [43].

## 3. Role of ERAP1 and ERAP2 on the RAS Imbalance in Patients with COVID-19

Following the interaction of SARS-CoV-2 with the host cell, the surface expression of ACE2 is reduced by endocytosis and cleaved by a proteolytic process [14] (Figure 3). This phenomenon appears to contribute to the pathogenesis of COVID-19, as the expression of ACE2 protects against acute lung injury [44,45,46]. Indeed, the downregulation of ACE2 leads to an excessive accumulation of Ang II and a consequent stimulation of the classical axis (ACE/Ang II/AT1R), leading to pulmonary injury, hematological alterations, and a hypertensive and hyper-inflammatory state [47] (Figure 4a). Consistent with this hypothesis, a cohort of 12 patients with COVID-19 showed an increase in circulating Ang II levels compared to healthy individuals, suggesting a possible link between the downregulation of tissue ACE2, a systemic imbalance of RAS, and an increased risk of multi-organ damage from SARS-CoV-2 infection [48]. Similarly, a cohort of 40 Italian COVID-19 patients showed an association between the degeneration of lung function and increases in both blood pressure and hypokalemia, both of which are attributable to reduced levels of ACE2 [49]. Based on these observations, the authors hypothesized that patients with COVID-19 are less efficient at counteracting the progressive activation of RAS following ACE2 downregulation and ALDO upregulation [49]. Lanza et al. described COVID-19 as the possible effect of RAS impairment since an imbalance between the ACE/Ang II/AT1R and ACE2/Ang-(1-7)/MasR axes results in multi-organ dysfunction and an uncontrolled inflammatory response [46]. The ongoing clinical trial on recombinant human ACE2 (rhACE2) as a treatment for patients with COVID-19 (APN01-COVID-19, NCT04335136) highlights the importance of RAS in the pathogenesis of this complex disease. In this context, it is plausible that the activities of ERAP1 and ERAP2 may play a role in mitigating the effects of ACE2 downregulation through the cleavage of circulating Ang II (Figure 4a). Based on these considerations, we speculate that the genetic variants of *ERAP1* and *ERAP2* linked to hypertension and autoimmune diseases [33,34,35,36,37], such as *ERAP1* rs30187, could enhance the stimulation of the ACE/Ang II/AT1R axis (Figure 4b), and thus help to explain the variety of courses, as well as the differential pattern of gender and ethnicity of COVID-19.

## 4. Conclusions

The mortality of severe COVID-19 patients is consistently associated with older age and male sex [3,4,5]. Hypertension, diabetes, and other traits related to obesity and cardiovascular disease have been associated with the most aggressive clinical outcome, but the roles of risk factors in determining disease severity have not yet been elucidated. To this end, several international consortia have been established to identify the rare or common monogenic variations that characterize individuals who develop severe forms of COVID-19 from those who, despite numerous exposures to SARS-CoV-2, do not become infected. Early results showed enrichment in the loss-of-function variants in 13 human loci involved in type I interferon (IFN)-mediated immunity to influenza infection in approximately 3.5% of previously healthy individuals facing severe disease [50]. One study reported that at least 10% of patients with life-threatening COVID-19 pneumonia have neutralizing autoantibodies against type I IFN [51]. Other authors have identified a 3p21.31 gene cluster as a genetic susceptibility locus in COVID-19 patients with respiratory failure and the potential involvement of the ABO blood group system [52,53,54]. These results demonstrate that genetic background is an important discriminating factor for disease outcome. However, by providing an explanation for disease severity in less than 15% of cases, they reinforce the need for further studies. In this context, given the role of ERAP enzymes in the regulation of RAS, hypertension, antigen processing, inflammatory disorders, and cytokines and NO release, we hypothesize that knowledge of *ERAP1* and *ERAP2* genotypes may be useful in identifying the appropriate therapy for each COVID-19 patient. In line with this hypothesis, a missense variant in the *ERAP2* gene has been linked to an increased risk of death in a small cohort of COVID-19 patients from the U.K. Biobank [55]. Furthermore, ERAP1 has been involved in the generation of peptides from the SARS-CoV2-S1 spike protein with the optimal length to bind MHC class I molecules, including the possible SARS-CoV2-protective HLA-B*15:03 allele [56,57].

Therefore, we can speculate that ERAP aminopeptidases may be actively involved in different processes during COVID-19 pathogenesis, with the first and foremost being the regulation of the RAS pathway and MHC class I antigen processing and presentation. Thus, during SARS-CoV-2 infection, the activities of ERAP1 and ERAP2 could help to compensate for the increase in Ang II levels resulting from ACE2 downregulation and, at the same time, ensure the efficiency of antigen processing.

## Figures and Tables

**Figure 1 ijms-22-01705-f001:**
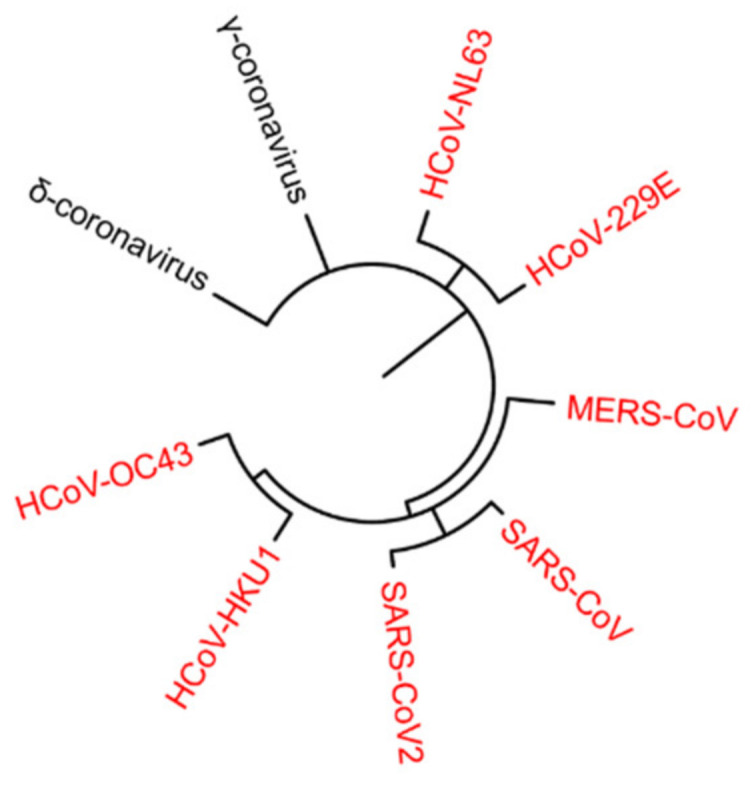
Schematic representation of the taxonomy of Coronaviridae. The species of the α- and β-coronaviruses that infect humans are in red. HCoV: human coronavirus, MERS-CoV: Middle East respiratory syndrome coronavirus, SARS-CoV: severe acute respiratory syndrome coronavirus.

**Figure 2 ijms-22-01705-f002:**
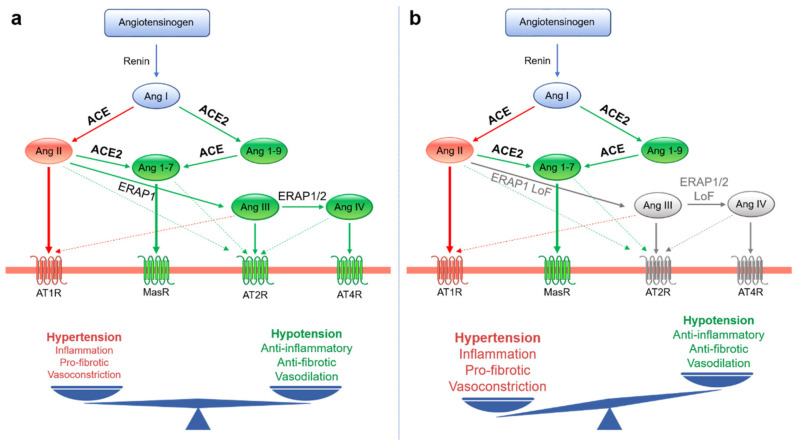
Schematic illustration of the renin-angiotensin system (RAS) showing the dual ACE- and ACE2-dependent pathways in the presence of functional (**a**) or dysfunctional (**b**) ERAP1 and ERAP2 enzymes. (**a**) In the classical ACE/Ang II/AT1R axis (red lines), Ang II binds its AT1R receptor, causing hypertension, inflammation, vasoconstriction, and fibrosis. The ACE2/Ang-(1-7)/MasR axis (green lines) counterbalances the harmful effects of the ACE/Ang II/AT1R axis. The cleavage of Ang substrates by ERAP1 and ERAP2 activities (green lines) is also shown. (**b**) Loss-of-function variants of ERAP aminopeptidases impair Ang III and Ang IV production, thus contributing to the increase of circulating Ang II levels and the resulting hypertension. Ang: angiotensin, ACE: angiotensin-converting enzyme, AT1R: angiotensin type-1 receptor, AT2R: angiotensin type-2 receptor, AT4R: angiotensin type-4 receptor, MasR: Mas receptor.

**Figure 3 ijms-22-01705-f003:**
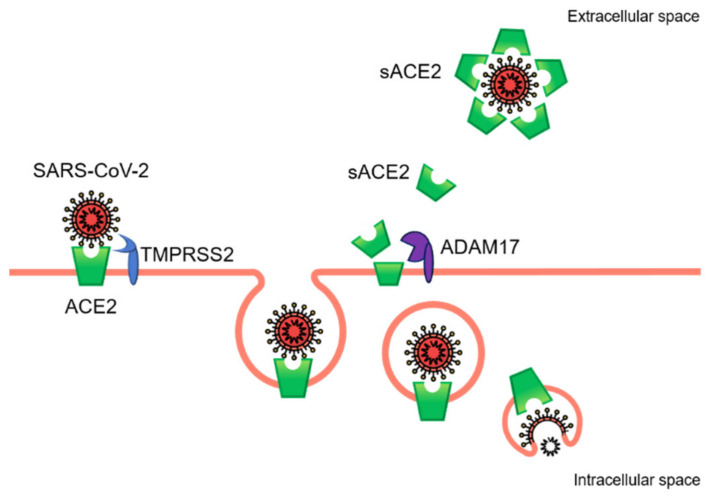
The fate of ACE2 during SARS-CoV-2 infection. After the binding of viral spike proteins to ACE2 and the S-protein priming by the transmembrane serine protease 2 (TMPRSS2), the viral particles are endocytosed. Acidification of the endosome leads to viral and cellular membrane fusion and the release of viral single-stranded RNA into the cytosol. The internalization of ACE2 contributes to the upregulation of ADAM metallopeptidase domain 17 (ADAM17), which cleaves the extracellular portion of ACE2 from the cell membrane. The soluble form of ACE2 binds the viral spike protein, contributing to attenuate the spread of the virus by impairing viral entry.

**Figure 4 ijms-22-01705-f004:**
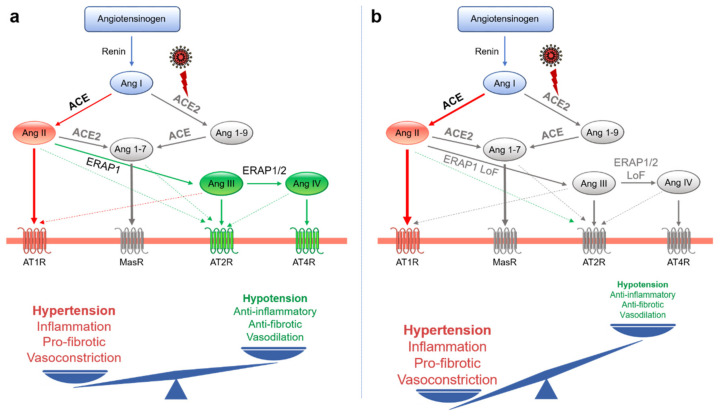
Schematic illustration of the renin-angiotensin system (RAS) in COVID-19 patients with functional (**a**) and dysfunctional (**b**) ERAP1 and ERAP2 enzymes. SARS-CoV-2 infection leads to ACE2 depletion, triggering an imbalance in the RAS, which allows for the accumulation of Ang II and a reduction of Ang-(1-9) and Ang-(1-7) substrates (**a**). This phenotype is further exacerbated in COVID-19 patients with dysfunctional ERAP1 and ERAP2 enzymes (**b**). The non-functional elements of the axes are shown in gray. LoF: loss of function.

**Table 1 ijms-22-01705-t001:** *ERAP1* and *ERAP2* polymorphisms associated with BP dysregulation.

Gene	SNP	Position	MA	MAF	Disease	Reference
*ERAP1*	rs469783	5:96785820	C	0.424	IHT	[39]
	rs10050860	5:96786506	T	0.167	IHT	[39]
	rs27772	5:96754272	G	0.378	BP progression	[39]
	rs27980	5:96762191	G	0.372	EHT	[40]
	rs17086651	5:96762144	C	0.070	EHT	[40]
	rs30187	5:96788627	T	0.363	AT1R inhibitor response; EHT; DBP and SPB increase;	[41,42,43]
*ERAP2*	rs2927615	5:96862499	A	0.183	IHT	[39]

BP: blood pressure, DBP: diastolic blood pressure, EHT: essential hypertension, IHT: incident hypertension, MA: minor allele, MAF: minor allele frequency, SBP: systolic blood pressure, SNP: single nucleotide polymorphism.

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
