# Peer review of "ERAP1 and ERAP2 Enzymes: A Protective Shield for RAS against COVID-19?"

_ijms, 2021, doi:10.3390/ijms22041705_

Round 1
Reviewer 1 Report
In the manuscript entitled “ERAP1 and ERAP2 enzymes: a protective shield for RAS against COVID-19?” the authors review current evident for the role of RAS in COVID-19 and based on known roles of ERAP1/ERAP2 on RAS, hypothesize that the dysfunctional status of these two enzymes may affect the severity and progression of COVID-19. The authors do a good job in reviewing the role of RAS in COVID as well the known association between ERAP1/ERAP2 and hypertension. However, the basis for generating the main hypothesis of the manuscript is rather problematic and constitutes a logical leap. While the authors name ERAP1/ERAP2 (line 68) as “key players” in the RAS axis, this is supported by 2 publications that are limited to showing in vitro cleavage of peptides and 1 showing in vivo effect of the mouse homologue of ERAP1 in mice (ERAAP) that displayed a small to moderate phenotype. It is not known if this phenomenon is translated to humans, especially given the relatively low homology between murine and human ERAP1 and the lack of ERAP2 in mice. Furthermore, that study focused on secreted ERAP1, and I am not aware if secreted ERAP2 has been demonstrated. Therefore, I don’t think that the notion that ERAP1 and ERAP2 are key players in the RAS axis is, at any level, established in the literature. Furthermore, in regards to lines 128-152, while some genome association studies have revealed association of ERAP1/ERAP2 polymorphisms with hypertension, these studies have been generally performed in small number of patients and thus have limited statistical significance. In contrast the association of ERAP1/ERAP2 with inflammatory autoimmunity in epistasis with HLA has been reproduced in multiple large studies. Thus, it is risky to form a hypothesis based on a less established function of these enzymes and not their well-established role in adaptive immunity through antigen presentation. It is odd that the authors do not explore a possible association between the function of enzymes that regulate immune responses and the progression of a viral infection – a much more straightforward hypothesis. Regardless, large genetic studies looking for genetic pre-disposition to severe COVID-19 have failed to identify ERAP1, although one study did identify ERAP2 (medRxiv 2020 Nov 9;2020.11.05.20226761). There are many genes involved in the RAS axis, that are more likely to influence COVID through that mechanism than ERAP1/ERAP2.
Some minor points:
Line 59, needs a comma after “fibrosis”
Line 60, this sentence is difficult to understand, please rephrase.
Line 101, please add a reference.
Lines 112-114: are these effects mediated by changes in antigen presentations? Please specify.
Lines 117: are the references 22,23 correct? They don’t seem to involve ERAP1/ERAP2
Author Response
Reviewer #1
In the manuscript entitled “ERAP1 and ERAP2 enzymes: a protective shield for RAS against COVID-19?” the authors review current evident for the role of RAS in COVID-19 and based on known roles of ERAP1/ERAP2 on RAS, hypothesize that the dysfunctional status of these two enzymes may affect the severity and progression of COVID-19. The authors do a good job in reviewing the role of RAS in COVID as well the known association between ERAP1/ERAP2 and hypertension.
We thank the Reviewer for all the comments aimed at improving our manuscript that we have addressed in the revised version. Below are the answers to the specific comments.
However, the basis for generating the main hypothesis of the manuscript is rather problematic and constitutes a logical leap. While the authors name ERAP1/ERAP2 (line 68) as “key players” in the RAS axis, this is supported by 2 publications that are limited to showing in vitro cleavage of peptides and 1 showing in vivo effect of the mouse homologue of ERAP1 in mice (ERAAP) that displayed a small to moderate phenotype. It is not known if this phenomenon is translated to humans, especially given the relatively low homology between murine and human ERAP1 and the lack of ERAP2 in mice. Furthermore, that study focused on secreted ERAP1, and I am not aware if secreted ERAP2 has been demonstrated. Therefore, I don’t think that the notion that ERAP1 and ERAP2 are key players in the RAS axis is, at any level, established in the literature.
As suggested by the Reviewer the sentence containing the reference of ERAPs to key players in the RAS, has been removed (Page 2, Line 76). Currently, 10 publications attest an involvement of ERAP enzymes in the degradation of angiotensin substrates and hypertension. Based on these data, we believe it is plausible to hypothesize a potential involvement of ERAP1 in the COVID-19 progression. Secreted ERAP2 has been demonstrated by Saulle et al Front Immunol 2019 (ref. 19).
Furthermore, in regards to lines 128-152, while some genome association studies have revealed association of ERAP1/ERAP2 polymorphisms with hypertension, these studies have been generally performed in small number of patients and thus have limited statistical significance. In contrast the association of ERAP1/ERAP2 with inflammatory autoimmunity in epistasis with HLA has been reproduced in multiple large studies. Thus, it is risky to form a hypothesis based on a less established function of these enzymes and not their well-established role in adaptive immunity through antigen presentation. It is odd that the authors do not explore a possible association between the function of enzymes that regulate immune responses and the progression of a viral infection – a much more straightforward hypothesis.
We are aware that the data on ERAP1 linkage to autoimmune disease are more impressive and therefore better known and disseminated than those on hypertension. However, this does not mean that the latter are not equally important, as they were statistically significant in the group of patients analyzed. As suggested by the Reviewer a possible association between the function of ERAP enzymes regulating immune responses and the progression of SARS-CoV2 infection has been formulated (Page 6 Line 230).
Regardless, large genetic studies looking for genetic pre-disposition to severe COVID-19 have failed to identify ERAP1, although one study did identify ERAP2 (medRxiv 2020 Nov 9;2020.11.05.20226761). There are many genes involved in the RAS axis, that are more likely to influence COVID through that mechanism than ERAP1/ERAP2.
The hypothesis formulated in the manuscript is intended to direct genetic research toward the study of ERAP1 enzymes for all the reasons discussed.
Some minor points:
Line 59, needs a comma after “fibrosis”.
As requested, a comma has been added after fibrosis (Page 2, line 68)
Line 60, this sentence is difficult to understand, please rephrase.
As requested, the sentence has been rephrased (Page 2, line 69)
Line 101, please add a reference.
As requested, a reference has been added (Page 3, line 106)
Lines 112-114: are these effects mediated by changes in antigen presentations? Please specify.
The results shown in Aldhamen’s manuscript do not appear to correlate a change in antigen presentation with the production of inflammatory cytokines by ERAP1 overexpressing and chemokines by ERAP1-overexpressing PBMC.
Lines 117: are the references 22, 23 correct? They don’t seem to involve ERAP1/ERAP2
Reference 12 (actual 17) refers to the role of ERAP1 and ERAP2 in regulating blood pressure by digesting plasma Ang II into Ang III and Ang IV, whereas references 22 and 23 (actual 28 and 29) refer to the Ang III and Ang IV ligands of Ang type 2 receptor (AT2R) and Ang type 4 receptor (AT4R), respectively. The sentence has been modified accordingly (Page 3, line 120).
Reviewer 2 Report
The review manuscript by D'Amico et al focuses on the role of ACE2 znd ERAP1/2 in the development of COVID. The presented hypothesis is interesting and well-presented. However, the authors focused only on ACE2 as SARS-CoV2 cell surface receptor. There were some reports that other cell surface proteins, e.g. CD147 could function as viral receptor as well. In my opinion presented manuscript would largely benefit from incorporating data regarding other receptors and their cellular functions. Furthermore, is there any link between CD147 (and its binding partners, intracellular-CD147-dependent proteins) and ACE2/ERAP?
Author Response
Reviewer #2
The review manuscript by D'Amico et al focuses on the role of ACE2 and ERAP1/2 in the development of COVID. The presented hypothesis is interesting and well-presented. However, the authors focused only on ACE2 as SARS-CoV2 cell surface receptor. There were some reports that other cell surface proteins, e.g. CD147 could function as viral receptor as well. In my opinion, presented manuscript would largely benefit from incorporating data regarding other receptors and their cellular functions. Furthermore, is there any link between CD147 (and its binding partners, intracellular-CD147-dependent proteins) and ACE2/ERAP?
We thank the Reviewer for appreciating our manuscript. As requested, the description of a new receptor used by SARS-CoV2 to enter host cells has been included (Page 2, line 53). No link was found between CD147 and ACE2/ERAP.
Reviewer 3 Report
The author provided a concise review on ERAPs in COVID-19. Overall, the manuscript is concise and highlighted the critical aspects of ERAP1 and ERAP2 functions in COVID-19 patients. Overall, this manuscript is well written and provide the new therapeutic approach in COVID-19. Certainly, this study is interesting and should attract a broad range of readership.
Comments
The authors mentioned that “mortality on severe COVID-19 patients is consistently associated with older age and male sex”. Is there any dataset available for male sex mortality rate ? Please discuss more about the mortality rate or include references about this data.
Please explain more about figure 2 and 4 in legend.
Typo error – Line 28
Author Response
Reviewer #3
The author provided a concise review on ERAPs in COVID-19. Overall, the manuscript is concise and highlighted the critical aspects of ERAP1 and ERAP2 functions in COVID-19 patients. Overall, this manuscript is well written and provide the new therapeutic approach in COVID-19. Certainly, this study is interesting and should attract a broad range of readership.
We would like to thank the Reviewer for considering our manuscript interesting.
Comments
The authors mentioned that “mortality on severe COVID-19 patients is consistently associated with older age and male sex”. Is there any dataset available for male sex mortality rate? Please discuss more about the mortality rate or include references about this data.
As requested, more details on the male sex mortality rate have been discussed and referenced (Page 1, line 35)
Please explain more about figure 2 and 4 in legend.
As requested, the legends of figures 2 and 4 have been more detailed
Typo error – Line 28
Line 28 has been corrected (Page 1 line 28)
Round 2
Reviewer 1 Report
minor comments:
Reference 20 is about ERAP1 only – add a reference for ERAP2 also
Line 119. While blood pressure regulation was demonstrated for ERAP1, none of the references for ERAP2 demonstrate this, only in vitro cleavage of peptides
